# Efficacy of Environmental Cleaning Protocol Featuring Real-Time Feedback with and without PX-UV in Reducing the Contamination of Gram-Negative Microorganisms on High-Touch Surfaces in Four Intensive Care Units in Thailand

**DOI:** 10.3390/antibiotics12030438

**Published:** 2023-02-22

**Authors:** Ornnicha Sathitakorn, Kittiya Jantarathaneewat, David J Weber, Piyaporn Apisarnthanarak, Sasinuch Rutjanawech, Anucha Apisarnthanarak

**Affiliations:** 1Division of Infectious Diseases, Faculty of Medicine, Thammasat University, Bangkok 12121, Thailand; 2Department of Pharmaceutical Care, Faculty of Pharmacy, Thammasat University, Bangkok 12121, Thailand; 3Research Group in Infectious Diseases Epidemiology and Prevention, Faculty of Medicine, Thammasat University, Bangkok 12121, Thailand; 4Gillings School of Global Public Health, University of North Carolina at Chapel Hill, Chapel Hill, NC 27599, USA; 5Division of Diagnostic Radiology, Department of Radiology, Faculty of Medicine Siriraj Hospital, Mahidol University, Bangkok 10700, Thailand

**Keywords:** ultraviolet disinfection, environmental cleaning, terminal cleaning, Gram-negative microorganisms, environmental disinfection

## Abstract

Environmental cleaning and disinfection practices have been shown to reduce microorganism bioburden in the healthcare environment. This study was performed in four intensive care units in Thailand. Five high-touch surfaces were sampled before and after terminal manual cleaning and disinfection, and after pulsed xenon UV (PX-UV). Five nursing station sites were collected on a weekly basis before and after terminal manual cleaning. There were 100 patient rooms—50 rooms in the intervention arm and 50 rooms in the control arm—plus 32 nursing station sites. In the intervention arm, rooms with positive Gram-negative microorganisms were reduced by 50% after terminal manual cleaning and disinfection (*p* = 0.04) and 100% after PX-UV disinfection (*p* < 0.001). On five nursing station sites, colony counts of Gram-negative contamination decreased by 100% (*p* < 0.001) in the intervention arm while decreasing by 65.2% (*p* = 0.03) in the control arm after terminal manual cleaning and disinfection. The in-room time use was 15.6 min per room. A PX-UV device significantly reduced the level of Gram-negative microorganisms on high-touch surfaces in intensive care units. The application of a PX-UV device was practical a in resource-limited setting without compromising cleaning and disinfection times.

## 1. Introduction

According to the United States Centers for Disease Control and Prevention (CDC), high-risk areas in intensive care units must receive terminal manual cleaning and disinfection after patient discharge or transfer to maintain a safe environment for future patients and to decrease healthcare-associated infections [1]. However, several studies have shown that manual cleaning and disinfection are often suboptimal [2,3,4] and demonstrated that only ~40% of surfaces were considered clean after terminal manual cleaning [5]. Monitoring and feedback on environmental cleaning practices have been shown to reduce microorganism bioburden in the healthcare environment [6,7]. Despite the reported up to 80% decrease in contamination by Gram-negative microorganisms after environmental feedback of cleaning practices in the United States of America [3], adherence to the protocol among hospitals in Thailand remains suboptimal [8]. Therefore, strategies to enhance environmental cleaning and disinfection are required in resource-limited settings.

The implementation of a pulsed xenon UV (PX-UV) light device (Xenex Healthcare Services, San Antonio, TX) disinfecting device in addition to standard cleaning and disinfection procedure may further decrease bioburden on contaminated surfaces and enhance the level of disinfection in the healthcare environment [9]. The germicidal UV used for surface disinfection is produced by a mercury vapor discharge tube and has a peak wavelength of 253.7 nm. The glass tube absorbs the ozone-producing short-wavelength UV [10]. UV-C, which uses a short wavelength of 250–280 nm, is considered the most lethal of wavelengths due to its capability of inactivating microorganisms as it is strongly absorbed into their nucleic acids [11]. UV-C irradiation is effective for reducing the environmental bioburden of methicillin-resistant *Staphylococcus aureus* (MRSA), vancomycin-resistant *Enterococcus* (VRE), *Clostridioides difficile* and multidrug-resistant (MDR) *Acinetobacter baumannii* on high-touch surfaces in hospital rooms [12,13,14,15]. Furthermore, studies have demonstrated that UV-C was associated with healthcare-associated infection reductions [16,17,18].

Several studies reported the potential use of UV light devices for reducing the incidence of Gram-positive bacteria (e.g., MRSA, *C. difficile*), but limited data are available on Gram-negative bacteria [9,19]. Further study is required to explore the effectiveness of PX-UV in reducing the bioburden of Gram-negative bacteria. Studies of PX-UV in resource-limited settings, which often face challenges in maintaining a high standard of environmental cleanliness, are also limited. The purpose of this study was to evaluate the efficacy of PX-UV in reducing the contamination level of Gram-negative microorganisms on high-touch surfaces in four intensive care units, featuring real-time feedback with and without PX-UV. The findings of this study have the potential to inform the development of more effective disinfection protocols in resource-limited settings to prevent the spread of harmful pathogens. This can be particularly important in environments where access to advanced disinfection technology is limited and there is a greater risk of disease transmission.

## 2. Results

There were a total of 100 patient rooms—50 rooms in the intervention arm (units A and B) and 50 rooms in the control arm (units C and D)—plus 32 nursing station sites—16 nursing station sites from the intervention arm and 16 nursing station sites from the control arm. The adhesion of bacteria to surfaces (as defined by mean colony count) varied from unit to unit (Table 1). Compared to the baseline, in the intervention arm, the number of rooms with positive Gram-negative microorganisms was reduced by 50% after terminal manual cleaning and disinfection [(20/50; 40%) vs. (10/50; 20%), *p* = 0.04] and reduced by 100% after PX-UV disinfection [(10/50; 20%) vs. (0/50; 0%), *p* < 0.001]. Compared to the baseline, in the control arm, the number of rooms with positive Gram-negative microorganisms was reduced by 50% after terminal manual cleaning and disinfection [(20/50; 40%) vs. (10/50; 20%), *p* = 0.04]. The predominant pathogen in the intervention arm and control arm was MDR-*Acinetobacter baumannii*, which was reduced by 33% after terminal manual cleaning and disinfection [(6/50; 12%) vs. (4/50; 8%), *p* = 0.73] in both arms, and reduced by 100% after PX-UV disinfection [(4/50; 8%) vs. (0/50; 0%), *p* = 0.11] in the intervention arm. The details of the median of the colony-forming unit per cm^2^ (CFU/cm^2^), including MDR-*Acinetobacter baumannii,* MDR-*Pseudomonas aeruginosa*, carbapenem-resistant *Enterobacterales* (CRE)-*Escherichia coli*, CRE-*Klebsiella pneumoniae*, extended-spectrum beta-lactamase (ESBL)*-Proteus mirabilis* and other Gram-negative microorganisms at the baseline, after standard terminal manual cleaning and after PX-UV disinfection, are shown in Table 1.

Compared to the baseline, in the intervention arm, the mean colony counts of Gram-negative microorganisms for all five sites were significantly reduced from 24.1 to 7.5 CFU/cm^2^ (68.9%, *p* = 0.002) after terminal manual cleaning and disinfection. Mean colony counts after PX-UV disinfection were decreased to 0 CFU/cm^2^ (100%, *p* < 0.001) for all five high-touch surfaces. Compared to the baseline, in the control arm, the mean colony counts of Gram-negative microorganisms for all five sites were significantly reduced from 31.8 to 7.2 CFU/cm^2^ (77.4%, *p* = 0.002) after terminal manual cleaning. On five nursing station sites that were routinely collected on a weekly basis, the colony counts of Gram-negative contamination in the intervention arm decreased from the baseline of 7.9 to 0 CFU/cm^2^ (100%, *p* < 0.001) after terminal manual cleaning and disinfection. In the control arm, the colony counts of Gram-negative contamination were reduced from the baseline of 6.9 to 2.4 CFU/cm^2^ (65.2%, *p* = 0.03) after terminal manual cleaning and disinfection. PX-UV disinfection was not used on the nursing stations. Details of the mean CFU/cm^2^ on five high-touch surfaces in the intervention and the control arm including the nursing station sites are shown in Table 2.

The compliance of the protocol was monitored by the XENEX robot. The rooms disinfected that met room protocol were defined by a minimum of two cycles of disinfections per patient room. In the intervention arm, the compliance of PX-UV disinfection was 80% (80/100) [unit A compliance level of 78% (39/50) and unit B compliance level of 86% (43/50)]. The in-room time use was 15.6 ± 5.6 (Mean ± SD) minutes per room. The transport to room time was 15.3 ± 1.5 (Mean ± SD) minutes per room. Details on the time for PX-UV device deployment are shown in Table 3.

## 3. Discussion

There are several notable findings in our study. First, our findings show that the level of Gram-negative microorganisms on five high-touch surfaces was reduced by terminal manual cleaning and significantly reduced after the implementation of the PX-UV device. Second, in the intervention arm, there were no Gram-negative microorganisms detected on any of the nursing stations after terminal manual cleaning and disinfection during the study, while certain Gram-negative microorganisms were present at the nursing stations in the control arm after terminal manual cleaning and disinfection. Third, adding the PX-UV device to the standard terminal manual cleaning and disinfection helped to achieve a 100% reduction in Gram-negative microorganisms after terminal manual cleaning and disinfection, without significantly increasing in-room time use.

Several studies reported the potential use of UV devices for Gram-positive bacteria. Nerandzic et al. [12] found that UV-C radiation is effective in reducing the frequency of MRSA and VRE cultures by 93% and of *C. difficile* cultures by 80%. Boyce et al. [14] reported that the automated mobile UV device can reduce aerobic colony counts from 90% before UV disinfection to 44% after UV disinfection. Morikane et al. [13] also reported a decrease in the baseline incidence of MRSA of about 28.5% in the intensive care unit in just one year after PX-UV device implementation. Rutala et al. [20] found that UV-C irradiation can reduce *C. difficile* spores by 99.8% within 50 min, where the number of samples positive for MRSA was reduced from 20.3% to 0.5% within 15 min. Weber et al. [18] reviewed multiple studies that evaluated the effectiveness of UV devices in inactivating microorganisms on test surfaces in a typical room setting, and the most common microorganisms were healthcare-associated pathogens including MRSA, VRE, and *C. difficile*. However, limited data are available on the potential use of PX-UV devices on Gram-negative bacteria. A study in Japan [13] showed that PX-UV device implementation decreased the baseline incidence of drug-resistant *Acinetobacter baumannii* by 63% in the intensive care unit after the first 6 months. In Thailand, where MDR-*Acinetobacter baumannii* is endemic, our study found that implementation of a PX-UV device reduced the number of rooms with positive Gram-negative microorganisms from the baseline of 40% to 20% after terminal cleaning and to 0% after PX-UV disinfection. Furthermore, the mean colony counts of Gram-negative microorganisms on all high-touch surfaces and all nursing station sites were also significantly reduced in the intervention arm within 4 months. To our knowledge, this is the first study that has evaluated the impact of patient room PX-UV implementation on nursing stations. Although there are no recommendations to routinely disinfect surfaces on nursing stations, it is interesting to find that in the unit where PX-UV was not used, the nursing station was found to be contaminated with the same organisms as those identified in the patient rooms. This suggests that the cleaning of patient rooms with PX-UV can have a positive impact on the microbial contamination of the nursing stations as well.

Standard terminal manual cleaning and disinfection is effective in reducing the transmission of pathogens via the environmental route; however, multiple studies have shown that terminal manual cleaning and disinfection is often suboptimal [2,5,21,22]. Carling et al. [21] reported that the overall thoroughness of terminal manual cleaning, expressed as a percentage of surfaces evaluated, was just 49%. A follow-up study by the same investigators suggested that 57.1% of the standardized sites were cleaned following the terminal manual cleaning and disinfection in the intensive care units [22]. Notably, terminal manual cleaning and disinfection can help reduce overall MDR microorganisms by approximately 40% in the United States [5]. Some techniques have been used for terminal disinfection including the use of hydrogen peroxide vapor and UV-light devices [2]. In our study, the level of Gram-negative microorganisms was reduced by approximately 69% after standard terminal manual cleaning and disinfection. Therefore, adding the PX-UV device to standard terminal manual cleaning and disinfection would help achieve a 100% reduction in MDR Gram-negative microorganisms after terminal cleaning, without compromising the disinfection time. The automated PX-UV device was easy to use. Typically, the terminal manual cleaning and disinfection time in Thailand hospitals took about 1 to 2 h [8,23]. In our hospital, standard terminal manual cleaning and disinfection took about 50 to 60 min, while the in-room time use of the PX-UV device took an average of 15 min. Therefore, adding the PX-UV device did not significantly increase the time taken to terminally disinfected intensive care unit rooms. The reported in-time use of PX-UV ranged from 15 to 20 min in the United States [15,20]. Limited data on PX-UV in-time use were available in Asia. Kitagawa et al. [24,25,26] reported that the median time of the in-room use of the PX-UV device ranged from 10 to 20 min per room, which is consistent with our study. Due to the fact that the PX-UV device does not physically clean a room (e.g., removing dust or stains), room cleaning must always precede disinfection.

There are some limitations in this study. First, this study was performed in a single center, which may limit the generalizability of our results to other settings. Second, this study measured the Gram-negative microorganism bioburden on high-touch surfaces in both inpatient rooms and nursing stations after terminal manual cleaning and disinfection, with and without PX-UV. Therefore, the impact of the PX-UV alone on the incidence of MDR Gram-negative microorganisms could not be evaluated. Finally, due to the fact that molecular sequencing was not performed, we cannot conclusively determine the true extent of the positive impact of cleaned and disinfected patient rooms on nursing station microbial contamination.

In conclusion, the PX-UV device significantly reduced the level of Gram-negative microorganisms on high-touch surfaces in the intensive care units and may have a positive impact on the microbial contamination of nursing stations. The application of the PX-UV device was practical in a resource-limited setting without compromising terminal cleaning and disinfection time. Further studies to evaluate the effect of PX-UV devices on the incidence of MDR Gram-negative microorganisms are needed.

## 4. Materials and Methods

### 4.1. Settings and Study Design

This study was performed in four intensive care units at Thammasat University Hospital (TUH), a 795-bed, tertiary care academic facility, from 1 September 2022 to 31 December 2022. The four intensive care units include unit A, a 10-bed medical unit, unit B, a 12-bed surgical unit, unit C, an 8-bed surgical unit, and unit D, a 15-bed medical unit. The purpose of this study was to evaluate the efficacy of PX-UV in reducing the contamination level of Gram-negative microorganisms on high-touch surfaces. All units used the same environmental protocol for daily and terminal manual cleaning and disinfection according to APSIC Guidelines for environmental cleaning and decontamination [27]. The environmental cleaning and disinfection practice includes general cleaning practice based on the risk of the patient population in the intensive care units, according to hospital infection control policies. These include daily cleaning and terminal manual cleaning and disinfection, which were routinely performed in all intensive care units, together with the addition of disinfection by chemical disinfectant, increased frequency of cleaning, auditing, and feedback to the housekeeper. Unit A and unit B were assigned to an environmental cleaning and disinfection protocol featuring the real-time feedback (using the LINE application, version 11 (Line Corporation, Tokyo, Japan)) of environmental culture results, together with PX-UV after terminal manual cleaning and disinfection, while unit C and unit D were assigned to an environmental cleaning protocol featuring the real-time feedback of environmental culture results without PX-UV.

### 4.2. Data Collection

The data collected included five sites of high-touch surfaces before terminal manual cleaning, after terminal manual cleaning, and after PX-UV (for unit A and B) and five nursing station sites before and after manual cleaning every week. According to previous studies, high-touch surfaces were defined by the data on the frequency of contact between healthcare personnel hands and specific environmental surfaces in a patient room [28,29]. High-touch surfaces are typically defined as any items that sustain more than three contacts per interaction in intensive care units [28,29]. We performed an observation in our intensive care units to measure the frequency of touching surfaces. High-touch surfaces were defined as any items that were touched by more than three contacts per interaction, and this includes the infusion pump, medication cart, bedside table, overbed table and vital sign screening. The level and type of Gram-negative microorganisms in the CFU/cm^2^ were recorded at all sites before terminal manual cleaning and disinfection, after terminal manual cleaning and disinfection, and after PX-UV. These high-touch surfaces include: (1) an infusion pump, (2) a medication cart, (3) a bedside table, (4) an overbed table, and (5) a vital sign screen. Five nursing station sites include: (1) a computer monitor, (2) a keyboard, (3) a nursing station chair, (4) a nursing station table, and (5) a walky-talky. The time for PX-UV device deployment was monitored, including transport to room time, retrieval time, waiting time to use, in-room use time and return to storage time. The compliance of the protocol monitored by the XENEX robot was recorded.

### 4.3. Protocol

After the initial sampling, environmental service personnel performed standard terminal manual cleaning and disinfection, using a chlorine disinfectant solution (Aurora chemical, Bangkok, Thailand), 200 parts per million (ppm) in concentration, to clean the floor using cotton-string mops. The five high-touch surfaces (e.g., infusion pump, medication cart, bedside table, overbed table, and vital sign screen) and other areas of the room (e.g., chair, light switch, ventilator and sink surround), including nursing station sites (e.g., computer monitor, keyboard, nursing station chair, nursing station table, and walky-talky), were cleaned and disinfected with quaternary compounds (DUAL QUATS) disinfectant wipes (Pose health care, Bangkok, Thailand). This hospital standard protocol was applied in all four intensive care units.

In units A and B, in addition to standard terminal manual cleaning and disinfection, the protocol was to use a PX-UV device containing a xenon gas flash bulb that operated at 2 Hz and emitted a broad spectrum of irradiation covering the UV-C spectrum of 200 to 280 nm. The device was wheeled into a strategic position located near high-touch surfaces in the room and set to irradiate for 5 to 7 min, as suggested by the manufacturer. Then, the device was wheeled to a second location in the room and run for an additional 5 to 7 min. The disinfection process takes approximately 15 to 20 min, which includes setup, irradiation cycles, and repositioning [9]. The compliance of the protocol was monitored by the XENEX robot. The rooms disinfected that met room protocol were defined by a minimum of two cycles of disinfection per patient room.

In each intensive care unit, five high-touch surfaces were selected for sampling before terminal manual cleaning, after terminal manual cleaning, and after PX-UV (for units A and B) (Appendix A). Five nursing station sites were routinely collected on a weekly basis. There was no PX-UV disinfection on the nursing station site according to the concern of being harmful to human skin. All intensive care unit beds from which a patient has been discharged or transferred after having occupied the room for a minimum of 48 h were enrolled. Environmental cleaning and disinfection practices were routinely monitored by assigned intensive care unit staff and feedback to the housekeeper in each intensive care unit, based on the hospital infection control policy.

In units A and B, five high-touch surfaces were sampled at three-time points: (1) immediately following patient discharge, (2) following a manual clean performed by cleaning services staff per hospital protocol, and (3) immediately after the completion of three PX-UV disinfection cycles. In units C and D, the same high-touch surfaces were sampled at the two-time points but without the PX-UV device.

### 4.4. Microbiology Method for Environmental Sampling

Surface samples were collected using the MacConkey contact plates (25 cm^2^ per plate) (Redipor, Cherwell Laboratories, Bicester, UK). The secure and package plates with cold packs were sent to the laboratory. The MacConkey contact plates were incubated at 36 °C ± 1 °C for 48 h. After that, the plate was removed from the incubator and the colony count was performed. For each plate, morphologically unique colonies were identified using the Matric Assisted Laser Desorption Ionization Time of Flight. Any colonies identified as *Acinetobacter* spp., *Pseudomonas* spp., *or Enterobacterales* had susceptibility testing performed by the Vitek using the Gram-negative susceptibility card AST-N288. Multidrug-resistant Gram-negative bacteria were defined as those resistant to three or more antimicrobial classes [30]. Plate count data were unblinded and added to an Excel spreadsheet. The efficacy of the environmental cleaning protocol was evaluated on the level of contamination of various Gram-negative microorganisms, including extended-spectrum beta-lactamase (ESBL)-producing *Enterobacterales*, carbapenem-resistant *Enterobacterales* (CRE)*, Acinetobacter baumannii, Pseudomonas aeruginosa* and other Gram-negative microorganisms (e.g., *Stenotrophomonas maltophilia*, *Aeromonas* spp., *Micrococcus* spp. and *Klebsiella* spp.).

### 4.5. Statistical Analysis

All analyses were performed using SPSS, version 26 software (IBM, Armonk, NY, USA). A chi-square test (two-tailed) was used to compare proportion for categorical variables, while an independent *t*-test was used for continuous variables. All *p* values were two-tailed, and *p* < 0.05 was considered statistically significant. This study was approved by the Ethics Committee No.3 of Thammasat University (protocol code MTU-EC-IM-6-078/65).

## Figures and Tables

**Table 1 antibiotics-12-00438-t001:** *Acinetobacter baumannii*, *Pseudomonas aeruginosa*, *Escherichia coli, Klebsiella pneumoniae, Proteus mirabilis* and other Gram-negative microorganisms at baseline, after standard terminal cleaning and after pulse xenon ultraviolet (PX-UV) disinfection.

	No. Room Positive/No. Room (%)					
	Median CFU/cm^2^ (Range, Min-Max)					
	MDR *A. baumannii*	MDR *P. aeruginosa*	*E. coli* (CRE)	*K. pneumoniae* (CRE)	*P. mirabilis* (ESBL)	Other Gram-Negative Microorganisms
Intervention arm (Units A and B) (n = 50 rooms)						
Baseline	6/50 (12)	2/50 (4)	0/50 (0)	4/50 (8)	2/50 (4)	6/50 (12)
	80 (10–200)	40 (30–50)	0 (0)	40 (10–70)	20 (10–30)	40 (10–200)
After terminal cleaning	4/50 (8)	0/50 (0)	0/50 (0)	2/50 (4)	0/50 (0)	4/50 (8)
	42 (10–100)	0 (0)	0 (0)	20 (10–30)	0 (0)	30 (10–50)
After PX-UV	0/50 (0)	0/50 (0)	0/50 (0)	0/50 (0)	0/50 (0)	0/50 (0)
	0 (0)	0 (0)	0 (0)	0 (0)	0 (0)	0 (0)
Control arm (Units C and D) (n = 50 rooms)						
Baseline	6/50 (12)	2/50 (4)	2/50 (4)	4/50 (8)	0/50 (0)	6/50 (12)
	60 (10–250)	60 (40–80)	20 (10–30)	40 (10–70)	0 (0)	60 (10–400)
After terminal cleaning	4/50 (8)	0/50 (0)	0/50 (0)	2/50 (4)	0/50 (0)	4/50 (8)
	30 (10–50)	0 (0)	0 (0)	20 (10–30)	0 (0)	30 (20–50)

CFU/cm^2^, colony-forming unit per cm^2^; MDR, multidrug-resistant. NB: Other Gram-negative microbe = *Stenotrophomonas maltophilia* (n = 2), *Aeromonas* spp. (n = 2), *Micrococcus* spp. (n = 2), *Klebsiella* spp. (n = 6).

**Table 2 antibiotics-12-00438-t002:** CFU/cm^2^ at baseline, after standard terminal cleaning and after PX-UV disinfection.

Surface Locations	Number of Rooms	Baseline	Standard Terminal Cleaning		PX-UV Disinfection	
		Mean CFU/cm^2^ (±SD)	Mean CFU/cm^2^ (±SD)		Mean CFU/cm^2^ (±SD)	
Intervention arm (Units A and B)						
Infusion pump	50	19.2 ± 20.1	1.9 ± 1.5	*p* = 0.005	0	*p* < 0.001
Medication cart	50	30.4 ± 15.1	11.6 ± 4.8	*p* = 0.004	0	*p* < 0.001
Bedside table	50	27.9 ± 11.5	9.4 ±12.2	*p* = 0.003	0	*p* < 0.001
Overbed table	50	31.2 ± 32.2	10.9 ± 9.4	*p* = 0.006	0	*p* < 0.001
Vital sign Screen	50	11.9 ± 11.5	3.9 ± 4.1	*p* = 0.005	0	*p* < 0.001
Mean Total sites	250	24.1 ± 12.4	7.5 ± 5.9	*p* = 0.002	0	*p* < 0.001
Intervention arm Nursing station						
Computer monitor	16	10.5 ± 8.9	0	*p* < 0.001	NA	
Keyboard	16	9.9 ± 8.4	0	*p* < 0.001	NA	
Nursing station chair	16	5.9 ± 5.1	0	*p* < 0.001	NA	
Nursing station table	16	2.4 ± 2.6	0	*p* < 0.001	NA	
Walky talky	16	3.9 ± 3.4	0	*p* < 0.001	NA	
Mean Total sites	80	7.9 ± 6.4	0	*p* < 0.001	NA	
Control arm (Units C and D)						
Infusion pump	50	30.1 ± 15.9	4.9 ± 2.5	*p* = 0.005	NA	
Medication cart	50	31.2 ± 10.4	2.4 ± 2.1	*p* = 0.004	NA	
Bedside table	50	40.4 ± 30.4	3.9 ± 2.9	*p* = 0.002	NA	
Overbed table	50	32.2 ± 20.4	10.4 ± 8.9	*p* = 0.003	NA	
Vital sign screen	50	24.9 ± 21.9	14.5 ± 12.9	*p* = 0.006	NA	
Mean Total sites	250	31.8 ± 24.2	7.2 ± 2.9	*p* = 0.004	NA	
Control arm Nursing station						
Computer monitor	16	12.9 ± 11.5	2.1 ± 2.1	*p* = 0.02	NA	
Keyboard	16	10.9 ± 9.4	3.5 ± 3.9	*p* = 0.04	NA	
Nursing station chair	16	4.9 ± 4.1	2.4 ± 2.1	*p* = 0.03	NA	
Nursing station table	16	1.6 ± 1.9	0	*p* = 0.001	NA	
Walky talky	16	2.9 ± 2.4	0	*p* = 0.002	NA	
Mean Total sites	80	6.9 ± 5.4	2.4 ± 2.2	*p* = 0.03	NA	

CFU/cm^2^, colony-forming unit per cm^2^; SD, standard deviation; NA, not available.

**Table 3 antibiotics-12-00438-t003:** Time for PX-UV device deployment.

Process	Mean ± SD (min)	Median (Min-Max) (min)
Transport to room time	15.3 ± 1.5	15 (5–25)
Retrieval time	6.4 ± 1.7	5 (3–10)
Waiting time to use	8.9 ± 2.0	10 (2–15)
In-room use time	15.6 ± 5.6	15 (7–25)
Return to storage time	8.5 ± 1.4	7 (3–11)
Total time	39.4 ± 2.4	37 (15–46)

SD, standard deviation.

## Data Availability

The data that support the findings of this study are available on request from the corresponding author. The data are not publicly available due to privacy or ethical restrictions.

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
