# Peer review of "Efficacy of Environmental Cleaning Protocol Featuring Real-Time Feedback with and without PX-UV in Reducing the Contamination of Gram-Negative Microorganisms on High-Touch Surfaces in Four Intensive Care Units in Thailand"

_antibiotics, 2023, doi:10.3390/antibiotics12030438_

Round 1
Reviewer 1 Report
The reviewer found the idea of the submitted manuscript titled ‘Efficacy of Environmental Cleaning Protocol Featuring Real-time Feedback with and without PX-UV in Reducing the Contamination of Gram-negative Microorganisms on High-touch Surfaces in Four Intensive Care Units in Thailand’ is interesting and can be published in Antibiotics. However, it is needed to be revised according to the following comments with a minor revision before the publication.
1. Authors used the acronym ‘PX-UV’, in the abstract part. However, they have not defined it.
2. How the authors decided on the High-touch surfaces. Can you include the references regarding this if any?
3. There is absolutely no pictorial evidence either in the data collection part or in the study of disinfection. Please include the relevant figures if possible.
4. Please consider the formatting correction during the revision of the manuscript.
Reviewer 2 Report
This manuscript is very interesting in the sense that it provides data from hospitals about bacteria contamination and its disinfection process. It gives the species found and the location. The authors could comment if some surfaces have more adhesion of some bacteria than others. The feasibility and efficacy of the PX-UV system is weel demonstrated. The authors might comment about other systems that may have similar performance. The authors might comment if there was a decrease in patient infection cases when using the PX-UV since this is the main goal. Once could say that there is 100% disinfection but is it necessary?
Reviewer 3 Report
Dear Authors,
I have reviewed your manuscript, and I am expressing my positive feedback. Your study is interesting for the Antibiotics journal readers, and the obtained results are promising. However, there is some space for improvement. Therefore, I am requesting revision according to the following comments:
· The abstract contains too many numerical results. Please rewrite it and mention just some main trends that were revealed by your study.
· The Introduction is too short. Please add some more relevant information to better explain the background of the problem you are addressing with your study.
· Materials and Methods section should be placed between Introduction and Results.
· I find the structure of the manuscript very difficult to follow. All results are in one chapter, while the complete discussion is in a separate chapter. It is much easier for readers to follow the content if the discussion immediately accompanies the results. In this way, readers have to constantly return to the previous chapter to check what are the results. Therefore, please combine your results and discussion chapters, and introduce corresponding sub-chapters for different types of results.
Once you address all of the above-mentioned comments, I will gladly review your manuscript again.
Best regards
